

# Defining Southern Ocean fronts using unsupervised classification

Simon D. A. Thomas[1,2], Daniel C. Jones[1], Anita Faul[1], Erik Mackie[3,1], and Etienne Pauthenet[4]

[1]British Antarctic Survey, NERC, UKRI, UK
[2]Department of Physics, University of Cambridge, UK
[3]Cambridge Zero, University of Cambridge, UK
[4]Sorbonne Universités, UPMC Université, LOCEAN-IPSL, Paris, France

**Correspondence:** Simon Thomas (sithom@bas.ac.uk)

**Abstract.** Oceanographic fronts are transitions between thermohaline structures with different characteristics. Such transitions are ubiquitous, and their locations and properties affect how the ocean operates as part of the global climate system. In the Southern Ocean, fronts have classically been defined using a small number of continuous, circumpolar features in sea surface height or dynamic height. Modern observational and theoretical developments are challenging and expanding this traditional
framework to accommodate a more complex view of fronts. Here we present a complementary new approach for calculating fronts using an unsupervised classification method called Gaussian mixture modelling and a novel inter-class parameter called the $I$-metric. The $I$-metric approach produces a probabilistic view of front location, emphasising the fact that the boundaries between water masses are not uniformly sharp across the entire Southern Ocean. The $I$-metric approach uses thermohaline information from a range of depth levels, making it more general than approaches that only use near-surface properties. We
train the statistical model on data from an observationally-constrained state estimate for more uniform spatial and temporal coverage. The probabilistic boundaries appear to be relatively sharp in the open ocean and somewhat diffuse near large topographic features, possibly highlighting the importance of topographically-induced mixing. For comparison with a more localised method, we use edge detection in principal component space and correlate the edges with surface velocities. The $I$-metric approach may prove to be a useful method for inter-model comparison, as it uses the thermohaline structure of those
models instead of tracking somewhat ad-hoc values of sea surface height and/or dynamic height, which can vary considerably between models. In addition, the general $I$-metric approach allows front definitions to shift with changing temperature and salinity structures, which may be useful for characterising fronts in a changing climate.

## 1 Introduction

The Southern Ocean (SO) is at the centre of the global thermohaline circulation, joining the Indian, Pacific, and Atlantic oceans into a single planetary-scale heat and carbon transport system (Marshall and Speer, 2012; Talley, 2013). In the SO, upwelling and downwelling branches of the overturning circulation transport water and tracers (e.g. heat, carbon) between the surface





and subsurface oceans (Sallee et al., 2010, 2012). The steeply tilted isopycnals associated with the overturning circulation also support the powerful Antarctic Circumpolar Current (ACC), with a mean combined barotropic and baroclinic volume transport of roughly $173.3 \pm 10.7$ Sv, driven by a combination of the westerly winds and air-sea buoyancy forcing (Rintoul et al., 2001; Morrison et al., 2015; Donohue et al., 2016). In part because of its unique structure, the SO is a critical regulator of global climate, having thus far absorbed more than 75% of the excess energy and 50% of the excess carbon added to the climate system from anthropogenic emissions (Mikaloff-Fletcher et al., 2006; Frolicher et al., 2015). As such, the thermohaline structure of the Southern Ocean may be considered an important climate system parameter, as it affects how heat and carbon are partitioned between the atmosphere and ocean.

Through decades of observational and theoretical effort, the global oceanographic community has curated a detailed theoretical understanding of the structure of the Southern Ocean. One of the hallmarks of this view is the presence of fronts, i.e. transitions in temperature, salinity, and/or biogeochemical properties (Deacon, 1937; Orsi et al., 1995). Although fronts are not identical to the sharp jets found in the SO, fronts and jets at the mesoscale share a close relationship partly due to thermal wind balance (Sokolov and Rintoul, 2002, 2009). Traditionally, oceanographers have defined SO fronts using a small number of continuous, circumpolar features that follow contours of sea surface height or dynamic height (Kim and Orsi, 2014). However, satellite altimetry shows that the ACC features a braided and meandering structure that is not necessarily reflected in the traditional, time-averaged view of fronts as continuous property contours (Chapman, 2017; Mackie, 2018). Using individual property contours to define fronts, for example contours of temperature or sea surface height, is somewhat limited by the fact that such contours do not always line up with the locations of strong gradients (Thompson et al., 2010; Thompson and Sallée, 2012; Graham et al., 2012; Chapman, 2017). In response to more detailed SO observations, the global oceanographic community has been developing a variety of new approaches for defining and tracking fronts in more application-specific ways (Chapman et al., 2020). For example, coastal applications and open ocean applications may benefit from conceptually different treatments of ocean fronts, which are characterised by different spatial and temporal scales. For a historical view and summary of advances in the area of front definition and detection, see the recent review article by Chapman et al. (2020).

In order to help us broaden our view of Southern Ocean fronts, we look to a branch of machine learning called unsupervised classification (also known as clustering). Broadly speaking, unsupervised classification attempts to identify sub-populations in data distributions that have not already been labelled or sorted. Although such methods have existed for decades, the amount of SO data has only in recent years become large enough for clustering approaches to be suitable; the application of unsupervised classification to oceanographic data is in its infancy. Several recent studies have used unsupervised classification to identify coherent regimes of thermohaline structure and the transitions between them, specifically in the North Atlantic (Maze et al., 2017), Southern Ocean (Jones et al., 2019), and Indian Sector of the Southern Ocean (Rosso et al., 2020). These methods have also been used to define coherent dynamical and biogeochemical regimes from depth-averaged ocean structure (Sonnewald et al., 2019; Le Bras et al., 2019; Jones and Ito, 2019). Recently, unsupervised classification has been used to define coherent ecological regimes from physical and biogeochemical data (Sonnewald et al., 2020). Researchers are also exploring potential connections between changes in class properties and large-scale climate phenomena. For example, a recent study tied evolution



in the longitudinal extent of an algorithmically-defined class to the onset of El Niño, suggesting that unsupervised classification methods could complement existing index-based assessments of large-scale climate modes (Houghton and Wilson, 2020).

Unsupervised classification does not use specific property contours to define boundaries between thermohaline structures, so it avoids one of the fundamental limitations of many traditional front definition approaches. Given the required information, unsupervised classification methods can use more detailed thermohaline data from throughout the water column to define classes and their boundaries. Across a given front, one might expect to find not only a transition in surface values but also a change in the thermohaline structure, as indicated by a change of profile class with latitude and/or longitude. In this work, we use an unsupervised classification technique called Gaussian Mixture Modelling (GMM), which attempts to represent sub-

populations in the data distribution using multi-dimensional Gaussian functions. Because GMM is a probabilistic method, in addition to automatically clustering the thermohaline profiles into classes, it returns for each data point a set of weights across the different classes. That is, it returns a probability distribution that can be exploited to define boundaries between coherent regimes in a novel way. In this paper, we propose that GMM can be used to represent the boundaries as "fuzzy" regions, which reflects the fact that not all transitions in the SO are uniformly sharp.

In Sect. 2, we introduce the observationally-constrained state estimate from which we draw our temperature and salinity data (Sect. 2.1), discuss principal component analysis (PCA) for dimensionality reduction (Sect. 2.2), and cover our application of GMM (Sect. 2.3). We then define the inter-class comparison metric (i.e. the $I$-metric) that we use to quantify water mass boundaries (Sect. 3.1). Next, we apply the $I$-metric to the reduced-dimension state estimate data (Sect. 3.2). For comparison, we contrast this method with a more local front-detection approach (Sect. 3.4). Finally, we discuss some caveats (Sect. 4) and

offer our summary and conclusions (Sect. 5).

## 2   State estimate data, PCA, and unsupervised classification

Our front identification method uses a combination of principal component analysis, unsupervised classification, and a new probabilistic metric to quantify the boundaries between coherent thermohaline structures. First, we describe the dataset that we used for developing and training our method.

### 80   2.1   The Southern Ocean State Estimate

We developed our method using the Biogeochemical Southern Ocean State Estimate (B-SOSE) (Verdy and Mazloff, 2017). B-SOSE is an observationally-constrained numerical simulation created using MITgcm [mitgcm.org] (Marshall et al., 1997a, b) and a suite of Southern Ocean observations, including Argo float data, ship track data, and satellite data. We chose to develop our method using a state estimate because such products offer (1) uniform coverage in latitude, longitude, and time and (2)

relatively high fidelity with respect to observations. In principle, our methods can be readily applied to any gridded temperature and salinity profile dataset. It may be possible to apply these methods to in-situ data as well, if the user addresses the problem of non-uniform spatial and temporal sampling. In this paper, we focus only on applications to gridded datasets.





To construct a state estimate like B-SOSE, researchers bring a numerical simulation into better consistency with an obser-
vational dataset using the 4DVAR method, which employs adjoint sensitivities to calculate the changes in initial conditions,
mixing parameters, and boundary conditions that are required to improve the agreement between the simulation and the dataset
(Stammer et al., 2002; Wunsch and Heimbach, 2007). Examples of other state estimates include the regional Southern Ocean
State Estimate (SOSE) and the global ECCOv4 state estimate (Mazloff et al., 2010; Forget et al., 2015).

The B-SOSE domain extends from the equator to 78°S, but we only use data south of 30°S to focus on the Southern Ocean
and to avoid the model boundary. It uses bathymetry and coastline based on Amante and Eakins (2009). B-SOSE solves the heat
and momentum equations using a third-order direct space and time advection scheme with a 1 hour timestep. The time-evolving
atmospheric boundary conditions use bulk formulae to solve for fluxes of heat, freshwater, and momentum, with six-hourly
atmospheric state variables as inputs (Large and Yeager, 2009; Dee et al., 2011). The state estimation process iteratively adjusts
the atmospheric state variables, oceanic initial conditions, and mixing parameters to improve model-data agreement. B-SOSE
uses dynamic sea ice (Losch et al., 2010; Fenty and Heimbach, 2013). For vertical mixing, it uses the GLL90 mixed layer
parameterization (Gaspar et al., 1990). It also uses horizontal and vertical viscosity and diffusivity. River runoff comes from
the product of Dai and Trenberth (2002) augmented with an estimate of Antarctic freshwater input from iceberg and ice sheet
melting (Hammond and Jones, 2016). It does not include mesoscale eddy parameterization, as this particular configuration
falls into the horizontal resolution range wherein mesoscale parameterization may actually worsen the representation of the
mesoscale (Hallberg, 2013). Because we are interested in quantifying physical, large-scale fronts, we only used monthly mean
temperature and salinity data. Also, because we are not interested in the surface seasonal cycle at present, we only used
temperature and salinity data between 300-2000 m, following Rosso et al. (2020). We used the whole period of iteration 106
of this state estimate, which covers January 2008 to December 2012. Some key properties of B-SOSE iteration 106 are listed
in Table 1. For further details, see Verdy and Mazloff (2017).

| Property | Value |
| --- | --- |
| State estimate iteration number | 106 |
| Horizontal resolution | 1/6° |
| Vertical resolution (variable) | 4.2 m to 400 m |
| Number of vertical levels | 52 |
| Output frequency | Monthly averaged |
| Horizontal viscosity | $10 \ \mathrm{m^2 s^{-1}}$ |
| Vertical viscosity | $10^{-3} \ \mathrm{m^2 s^{-1}}$ |
| Horizontal diffusivity | $10 \ \mathrm{m^2 s^{-1}}$ |
| Vertical diffusivity | $10^{-4} \ \mathrm{m^2 s^{-1}}$ |

**Table 1.** Selected properties of B-SOSE iteration 106. Output frequency refers to the output selected for this study.



## 2.2 Principal component analysis

Each vertical profile of temperature (T) or salinity (S) is comprised of 52 depth levels, so at every B-SOSE grid cell and output month, there are 104 values to be considered. Values close to each other in the water column are correlated to some degree. Therefore, we do not necessarily need values of T and S at every pressure level to capture most of the variability with depth. Principal component analysis (PCA) identifies the combinations of values that capture most of the variability with depth in the dataset, thus revealing potential physical structures that may be useful for understanding the stratification of the SO. We

choose the number of principal components such that the information carried in the remaining variability is negligible for our purposes. This data reduction also improves the speed and efficiency of the computational algorithms.

Following Rosso et al. (2020), we only keep values between 300 m - 2000 m to exclude most of the surface seasonal variability from the dataset. Because the data is spaced on an irregular grid in the vertical direction, we first interpolate the temperature and salinity profiles onto a regular grid with 10 m cells in the vertical. Following Pauthenet et al. (2017), at each

grid cell and time we combine the temperature and salinity profiles into a single vector. We normalise each depth level for both temperature and salinity separately. That is, we standardise the temperature values using the distribution of temperatures at the same depth level, and we standardise the salinity values using the distribution of salinities at the same depth level. This is a slightly different approach from Pauthenet et al. (2017), in which the authors standardise across the entire dataset. We found that for the work shown in this paper, the choice of normalisation approach does not make a large difference in the results

(not shown). After normalisation, we carry out PCA decomposition (sometimes called PCA expansion). We keep the first three principal components (PCs), which together statistically explain 98% of the variability across the thermohaline dataset.

The coefficients associated with the PC1 indicate a broad division between polar, high-latitude Southern Ocean waters and the subtropics (Fig. 1(a)). The most negative PC1 coefficients are found in the Weddell Gyre, and we also see the imprints of the South Pacific Gyre and the ACC (Vernet et al., 2019). The coefficients of PC2 bear the imprint of the ACC and of its

northward flow along the East Pacific basin (Fig. 1(b)). This northward flow is associated with the formation and export of Subantarctic Mode Water and Antarctic Intermediate Water (Iudicone et al., 2007; Sallee et al., 2010; Jones et al., 2016). PC2 also has the imprint of the Agulhas Current around South Africa. Finally, PC3 has strong negative values in the Weddell Gyre and over most of the Pacific, with a band of circumpolar positive values that somewhat mirrors the southward drift of the ACC when considered from west to east. The spatial structure of PC1 and PC2 are largely consistent with those of Pauthenet et al.

(2017), but the structure of PC3 is somewhat different from theirs, particularly in the subtropics. These differences are possibly a result of our choice of a different depth range. Given that PC3 explains a small fraction of the variability (7% of the variance explained), we do not expect these differences to impact our results.

After we perform dimensionality reduction, each monthly output at each model grid cell in latitude and longitude is represented using the first three coefficients of the PC expansion. The three PC values contain combined information about both

temperature and salinity, simplifying our analysis. This approach defines an abstract three-dimensional space in which we can perform unsupervised classification. In typical machine learning terminology, this abstract three-dimensional space can be called the "feature space", in which each PC axis is a "feature". To be explicit, we can say: each combined temperature-salinity





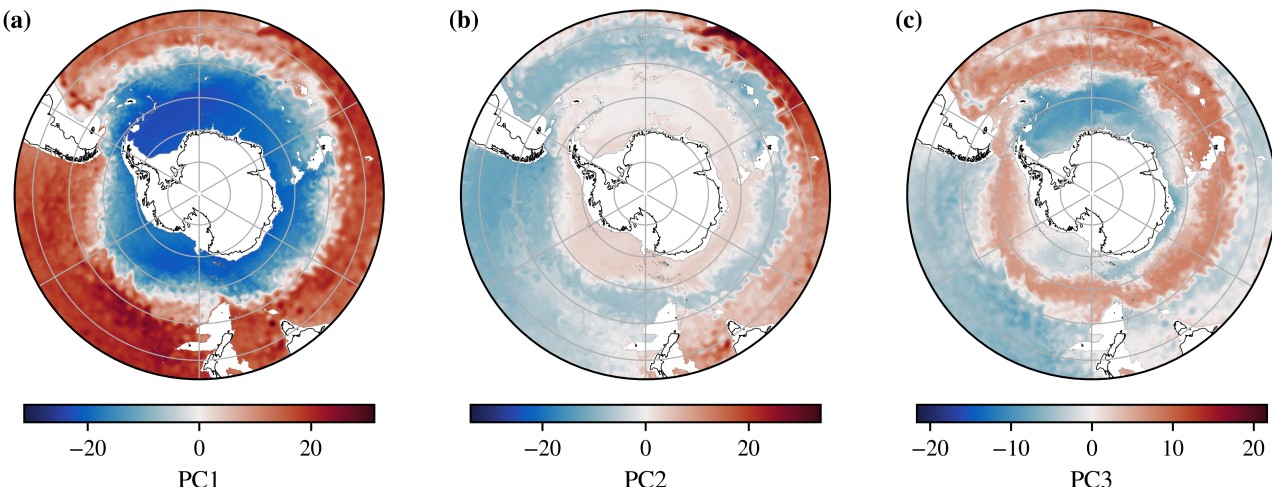

**Figure 1.** Each combined temperature and salinity profile can be approximated using a three-term PC expansion. Above are monthly mean coefficients of the PC expansion from June 2011. In order to limit the influence of seasonal variability, we use temperature and salinity profiles between 300m-2000 m. The first three PCs explain (a) 75%, (b) 16%, and (c) 7% of the variance respectively, together explaining a total of 98% of the variability. The white space represents bathymetry shallower than 2000 m, and its boundary is marked by a grey line.

profile in latitude, longitude, and time is represented by a three-dimensional vector of PC values. Each three-dimensional PC vector derived from B-SOSE is an "observation". In the next section, we use unsupervised classification to identify sub-

populations in the three-dimensional distribution of PC values.

## 2.3  Gaussian mixture modelling

Unsupervised classification attempts to identify sub-populations within a data distribution, without the assistance of any prede-fined labels. In our application, we attempt to identify data sub-populations in the abstract three-dimensional space defined by the PC coefficients (i.e. the "feature space"). Here we use GMM, an algorithm which attempts to fit a set of multi-dimensional

Gaussian functions to the data by iteratively adjusting the means and covariances of the Gaussians (McLachlan and Basford (1988), see appendix B for more detail). This method has recently been used to classify Argo temperature profiles in the top 2 km of the North Atlantic Ocean and the Southern Ocean (Maze et al., 2017; Jones et al., 2019). GMM is well-suited to ocean applications because it offers a probabilistic measure of classification in the form of posterior probabilities, which is useful when working with a highly correlated dataset. Because GMM-derived clusters will likely feature some overlap due to the

highly correlated nature of ocean data, such posterior probabilities offer an important complement to the GMM-derived class labels. In this application, we use the posterior probabilities to define coherent thermohaline regimes and their boundaries.

The GMM method attempts to represent the underlying data distribution using a set of $K$ Gaussian functions in $D$ dimen-sions (in our case $D = 3$):




$$\mathcal{N}\left(\boldsymbol{x}; \boldsymbol{\mu}_k, \Sigma_k\right) = \frac{\exp\left[-\frac{1}{2}\left(\boldsymbol{x} - \boldsymbol{\mu}_k\right)^T \left(\Sigma_k{}^{-1}\right)\left(\boldsymbol{x} - \boldsymbol{\mu}_k\right)\right]}{\sqrt{(2\pi)^D \|\Sigma_k\|}}, \tag{1}$$

where $\boldsymbol{x} \in \mathbb{R}^{D \times 1}$ is a vector in the PC space, $\boldsymbol{\mu} \in \mathbb{R}^{D \times 1}$ is the center of the Gaussian distribution expressed in vector form, $\Sigma_k \in \mathbb{R}^{D \times D}$ is the covariance matrix, and $|\Sigma_k|$ is its determinant. The covariance matrix determines the orientation of the Gaussian ellipsoids in PC space. We model the dataset, in the statistical sense of representing the dataset using a probability distribution, as a weighted sum of Gaussians:

$$\mathbb{P}(\boldsymbol{x}) \approx \sum_{k=1}^{K} \lambda_k \, \mathcal{N}\left(\boldsymbol{x} \, ; \, \boldsymbol{\mu}_k \, , \, \Sigma_k\right), \tag{2}$$

where $\lambda_k$ is the weight associated with the $k$-th Gaussian. The process of fitting the GMM uses expectation maximisation (EM), which consists of iteratively adjusting $\lambda_k$, $\boldsymbol{\mu}_k$, and $\Sigma_k$ to decrease the model-data misfit. For additional details, see Appendix B.

Once the weights, means, and covariances are fixed, each data vector $\boldsymbol{x}$ is associated with a probability distribution across all of the $K$ classes. This distribution is the set of likelihoods that the data vector belongs to any particular class, and the probabilities sum to one. GMM assigns each data vector to the class with the maximum posterior probability. We will now use this distribution to define an inter-class metric, which gives us a novel perspective on fronts as transitions in thermohaline structures.

## 3 The inter-class comparison metric (the *I*-metric)

First, we examine the structure of our profile data in PC space and introduce the *I*-metric for identifying boundaries between coherent hydrographic regimes (Sect. 3.1). Next, we examine the *I*-metric in both a monthly averaged and multi-year averaged view in latitude-longitude space, and we explore the class structure in more detail by examining the associated coherent regions and vertical profile types (Sect. 3.2). Following that, we compare our results with a local edge detection method (Sect. 3.4).

### 3.1 Defining the *I*-metric

For each combined temperature-salinity profile, GMM returns a probability distribution across all of the $K$ classes. This distribution is called the *posterior probability* distribution, and it quantifies the probability that a particular profile is in a particular class. If the posterior probability is close to 1.0 for class $k$ and very small for the other classes, then within the context of the Gaussian statistical model (i.e. GMM), the classification of the profile into class $k$ is unambiguous and clear. However, if the posterior probability is close in value for the two classes with maximum probabilities, then the classification is ambiguous and less clear. With this in mind, we can use the difference between the maximum probability and the second-highest probability to quantify how clearly the profile has been classified. If the classification is unambiguous, then the profile is less likely to be associated with a boundary between coherent thermohaline regimes. If the classification is ambiguous, then




the profile is more likely to be associated with a boundary. With this in mind, we propose a probabilistic inter-class comparison metric of the form:

$$I(\mathbf{x_n}) = 1 - \left[ \mathbb{P}(c = c_k)_{\mathrm{max}} - \mathbb{P}(c = c_l)_{\mathrm{runner-up}} \right], \qquad (3)$$

where $\boldsymbol{x}_n$ is the $n^{\mathrm{th}}$ profile's PC values and $\mathbb{P}(c = c_k)_{\mathrm{max}}$ is the maximum posterior probability that GMM has assigned the $n^{\mathrm{th}}$ profile as belonging to class $k$. The term $\mathbb{P}(c = c_l)_{\mathrm{runner-up}}$ is the second-highest posterior probability. If the difference between the maximum and runner-up posterior probabilities is close to one, then $I$ is small, indicating that the profile is not likely to be associated with a boundary between thermohaline regimes. If the difference between the maximum and runner-up posterior probabilities is small, then $I$ is close to one, indicating that the profile is likely to be associated with a boundary between different thermohaline regimes. The $I$-metric offers an alternative method for defining boundaries as fuzzy transitions between coherent regimes. In general, some regions will feature sharp transitions across boundaries, whereas other regions will feature more gradual transitions. The relative sharpness of a transition is influenced by the processes that form, mix, and destroy water masses. In contrast with approaches that define fronts as sharp transitions located along property contours or local gradients, the $I$-metric approach allows for a wider variety of transition types between regimes.

In our $I$-metric application, GMM clusters the profiles in feature space (Fig. 2(a)). The structure of the data shown in PC space is broadly consistent with that found in other studies (e.g. Pauthenet et al. (2017, 2018, 2019)). The data distribution is reasonably well represented by a linear combination of multi-dimensional Gaussian functions (Fig. 2). The $I$-metric values indicate transition regions between classes, where the class labelling is relatively ambiguous (Fig. 2(b)). We choose $K = 5$ to represent the general, large-scale pattern of the data; we explore the sensitivity of our results to $K$ in Sect. 4.4. In the next section, we examine the $I$-metric and class structure in physical space.

## 3.2 Geographic view of the *I*-metric

The $I$-metric viewed in latitude-longitude space illustrates the rich variety of transition types found in the Southern Ocean (Fig. 3). In all sectors of the SO, we see sharp transitions where the regions of high $I$ values are narrow and more gradual transitions where the regions of high $I$ values are more spread out. Some features are circumpolar, in consistency with the view of SO fronts as continuous lines that encircle Antarctica. However, we also see regions where the continuity and circumpolar nature of the fronts is not as clear, suggesting that a broader view may be appropriate (Chapman et al., 2020). The fronts are not uniformly sharp across all longitudes; for example, the northernmost transition is broad and gradual in the Atlantic sector, sharp in the Indian sector, and relatively broad in the Pacific sector. The southernmost band of high $I$-metric values is relatively sharp in the Atlantic sector, becoming increasingly broad as we follow it into the Indian and Pacific sectors. In the Pacific sector, it extends into an especially broad region in the Amundsen Sea, in consistency with the intersection of the classically-defined southern boundary (SBdy) with the Antarctic continent (Kim and Orsi, 2014). Upstream of Kerguelen Plateau, there is a region where the $I$-metric is spread out and diffuse between classes 2 and 3; this region also features a standing meander associated with enhanced eddy kinetic energy (Frenger et al., 2015; Siegelman et al., 2019). The enhanced mesoscale eddy kinetic energy associated with the meander is consistent with increased lateral mixing and the spread out pattern in the $I$-metric found in the





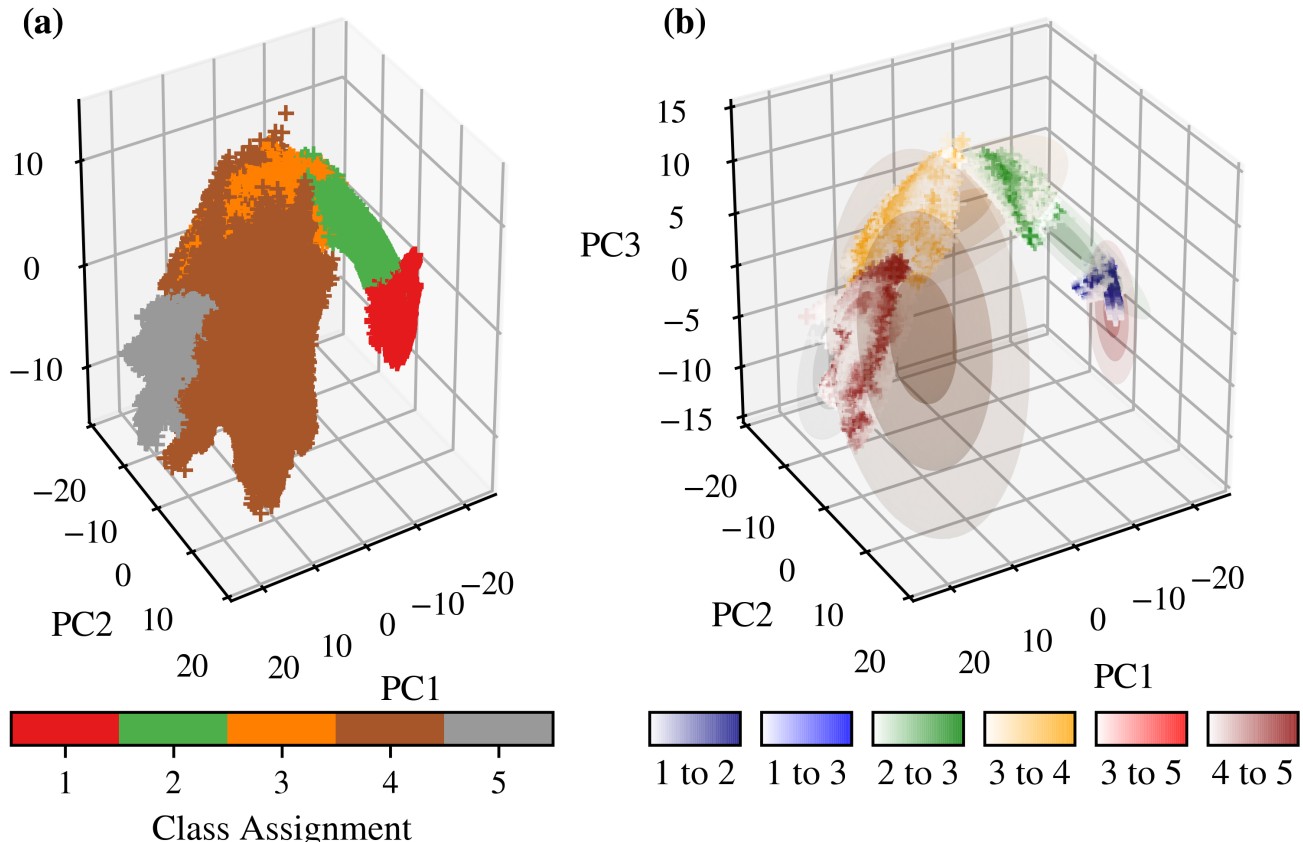

**Figure 2.** (a) The classification analysis takes place in the abstract PC space. Each point represents a 3D vector of principal component values that describe a single combined temperature and salinity profile. The three axes are the three principal components. Class assignments are indicated by the colours. (b) The *I*-metric highlights transitions between classes in the abstract PC space. The Gaussian ellipsoids of the GMM are shown in red, and the *I*-metric values associated with each point are shown using six different colour scales. Each colour scale corresponds to a particular transition between classes. Points with low *I*-metric values are not shown. The above is a subset of data taken from 12 months of monthly averaged B-SOSE data, inclusively between August 2011 and July 2012.

same region. Closer to the Antarctic continent, we also see the imprints of both the Weddell Gyre and the Ross Gyre, in regions of coherent structures with low *I*-metric values, in part enforced by the gyre circulation.

     The monthly mean *I*-metric (Fig. 3(a)) also highlights individual ring-like eddies; although these features are not typically considered fronts, they are small-scale transition regions between different hydrographic structures. We do expect the *I*-metric to be non-zero across these features. The monthly view also features mesoscale meanders, highlighting the detailed structure of the SO, which is partly a result of the energetic mesoscale eddy field. The *I*-metric does feature some month-to-month






variability; in some locations the fronts meander in their north-south extent, and in others they are relatively stationary, likely due to bathymetric constraints (see animations in Thomas (2021), in the "gifs" directory).

By averaging the four years worth of monthly means, we obtain a map of the climatological $I$-metric, which is averaged over many eddy lifetimes (Fig. 3(b)). Comparing an example monthly field with the climatological field, we can examine the
imprint of eddy spatial variability and the meandering of the fronts on the $I$-metric pattern. Most of our observations about the metric are unchanged by this averaging; we identify three roughly circumpolar bands of high $I$-metric values, with significant spatial variability and some overlap. The three bands are fairly distinct in the Atlantic sector, with the northernmost transition being the broadest. Upstream of Kerguelen plateau, the two northernmost bands become somewhat hard to distinguish. This is possibly a consequence of the eddy mixing and upwelling hotspot in that region, which tends to spread out hydrographic
features in latitude-longitude space, increasing the degree of spatial correlation found there. Upstream of Kerguelen plateau, the Polar Front features strong seasonal variability (Pauthenet et al., 2018). Note that the $I$-metric band aligned roughly with the Polar Front only passes south of the plateau (e.g. south of Heard Island), in consistency with other studies of the subsurface component of the Polar Front (e.g. Pauthenet et al. (2018)).

The three bands of higher $I$-metric values are distinct downstream of the Kerguelen plateau in the Indian sector; notably,
the southernmost band features especially high $I$-metric values in this sector. This pattern is associated with the transition between the Antarctic Circumpolar Current and the Antarctic Slope Current (ASC), which tend to flow in opposite directions (Thompson et al., 2018; Pauthenet et al., 2021). In the Pacific sector, we see the southernmost band turn into the Amundsen Sea and intersect with the Antarctic continental slope, spreading out in a diffuse region that is consistent with the behaviour of the southernmost extent of the ACC, the eastern boundary of the Ross Gyre, and the eastward shelf circulation along the
West Antarctic Peninsula (Nakayama et al., 2018). In this same sector, the northernmost band features the signature of two northward export pathways of Subantarctic Mode Water and Antarctic Intermediate Water, namely in the central and eastern Pacific (Iudicone et al., 2007; Sallee et al., 2010, 2012; Jones et al., 2016). These export pathways are influenced by the basin-scale stratification and the structure of the South Pacific Gyre.

### 3.3 Properties of the thermohaline regimes

In order to better understand the coherent thermohaline regimes underlying our $I$-metric results, we examine their lateral extents and their vertical properties. Despite not being given any latitude or longitude information, the underlying GMM captures several coherent, large-scale features of Southern Ocean thermohaline structure (Fig. 4(a)). Class 1 contains the coldest waters in the SO, covering both the Weddell and Ross gyres near Antarctica. The mean profile in this class features cold temperatures that are nearly uniform with depth; in general, they are salt stratified in that the near-surface waters are fresher than the
subsurface waters, ensuring that the density profile is stable overall (Fig. 5). The boundary between class 1 and class 2 broadly lies between the classically-defined Southern ACC front (SACCF) and the Southern Boundary (SBDY) (Kim and Orsi, 2014), including the turn of the SBDY towards almost being perpendicular with the Antarctic continent in the Pacific sector of the SO. Class 2 is circumpolar, with excursions into the Amundsen Sea and the area just south of Kerguelen Plateau. It features salt stabilisation, with a fresh layer near 300 m (Fig. 5). Class 3 is also circumpolar, with a northward excursion in the Atlantic




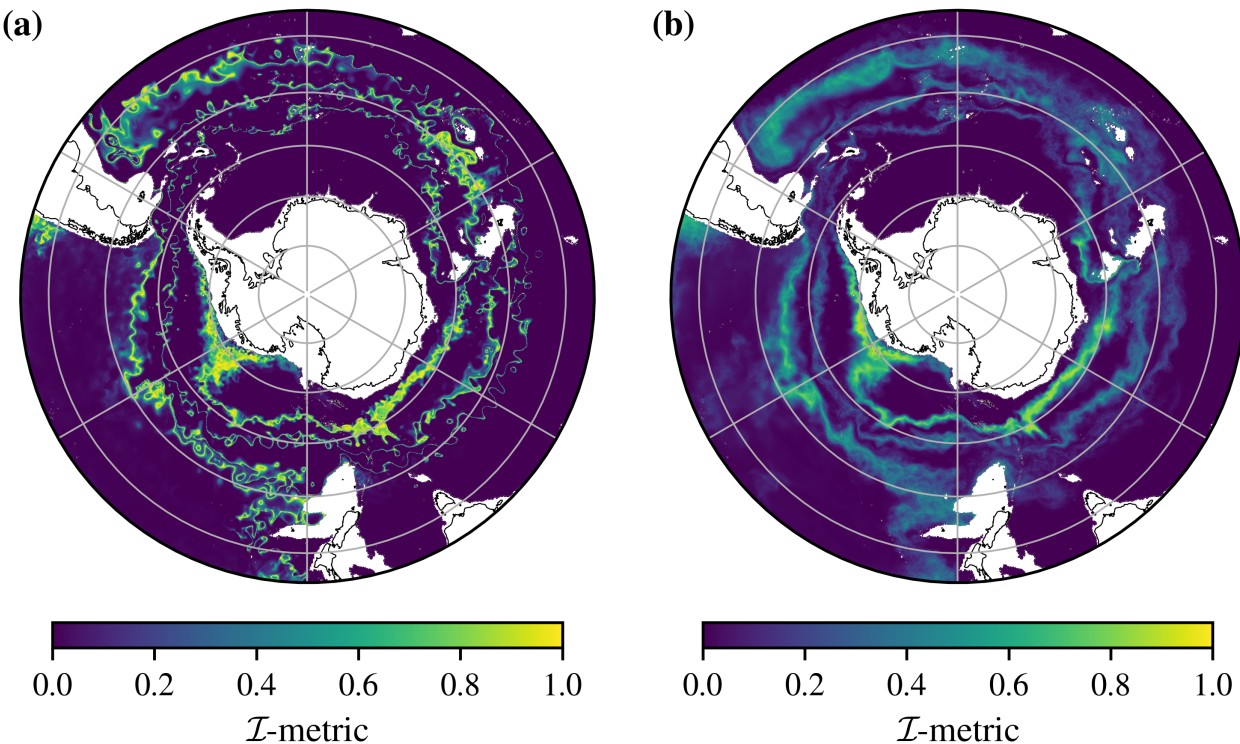

**Figure 3.** The *I*-metric highlights transitions between coherent thermohaline regimes in geographical space. (a) is the *I*-metric for a single month (April 2012) and (b) for the time average of the BSOSE-i106 dataset (60 months). Latitude lines are shown between 80°S and 40°S every 10°, and longitude lines are shown every 60°. Animations showing month-to-month and interannual variability are available in the software release (Thomas, 2021).

sector. The boundary between classes 2 and 3 roughly follows the Polar Front (PF), separating the colder, fresher Antarctic waters from the warmer, saltier subtropical waters further north. Finally, there are two subtropical classes; class 4 represents the Atlantic and Indian sectors of the subtropics, and class 5 represents the large-scale South Pacific Gyre. The boundary between classes 3 and 4 roughly aligns with the Subantarctic Front (SAF), particularly over large portions of the Indian and Pacific sectors (Fig. 4). The mean of class 5 has a salinity minimum around 700 m, corresponding to the presence of the Antarctic
Intermediate Water layer (Iudicone et al., 2007).





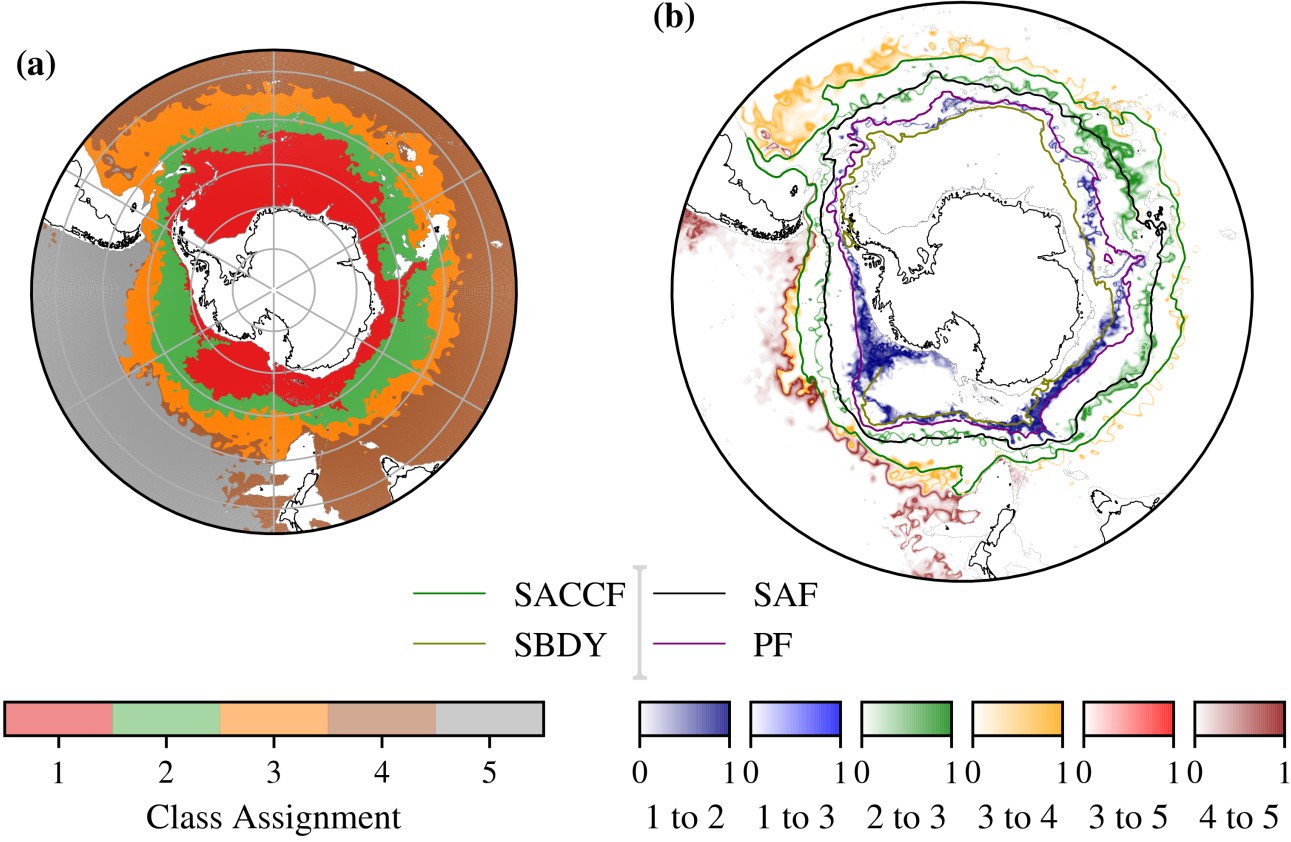

**Figure 4.** (a) The cluster assignments with $K = 5$ and (b) I-metric for specific class transitions. This view highlights the transitions between specific classes; the transitions have some similarities with respect to the altimetric fronts shown overlaid from Kim and Orsi (2014) (SBDY: Southern boundary, SACCF: Southern ACC front, PF: Polar front, SAF: Sub-Antarctic front). Data from June 2011 as a representative month.

### 3.4 An edge detection approach towards identifying fronts

For comparison with the GMM method, which uses properties of an entire training dataset to detect changes in thermohaline structure, we use a more local front detection method implimented by Hjelmervik and Hjelmervik (2019) in the North Atlantic. This method, called the Sobel method, directly examines spatial gradients in the principal component fields using a Sobel operator (Sobel and Feldman, 1968). To do this, the PCs of each grid point are placed onto a rectangular grid with the same spacing as the data sampling, where points without data are masked. The strength of an edge at a point is found by the two dimensional convolution (represented by *) of the gridded PCs and the following two matrices. In the $x$ direction the Sobel





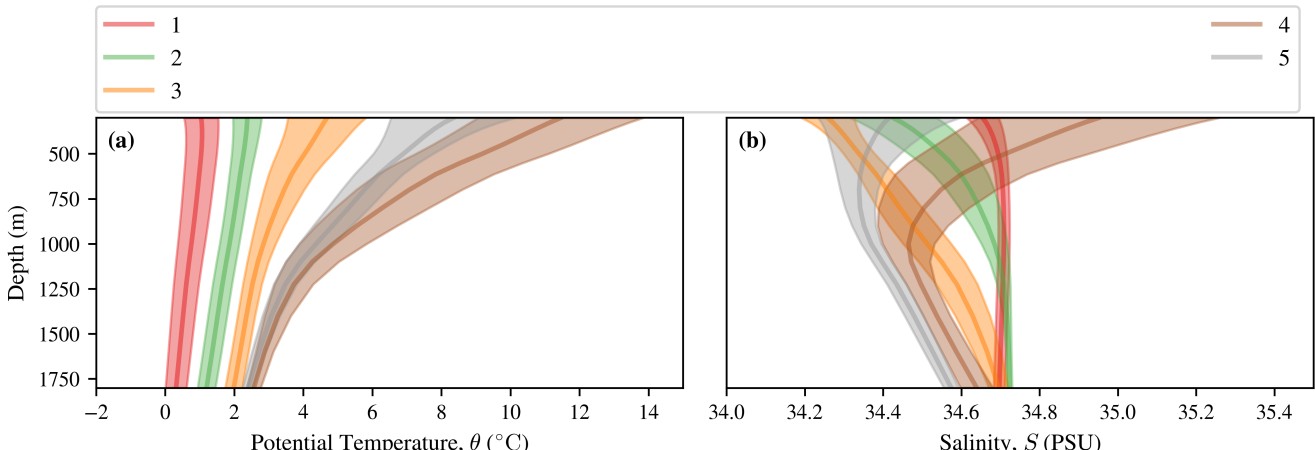

**Figure 5.** Profiles of the five GMM clusters between 300 m and 1800 m in (a) temperature and (b) salinity, for the training data, with one standard deviation envelope either side plotted.

operator is

$$G_x = \begin{bmatrix} 1 & 0 & -1 \\ 2 & 0 & -2 \\ 1 & 0 & -1 \end{bmatrix}, \tag{4}$$

and in the $y$ direction the Sobel operator is

$$G_y = \begin{bmatrix} 1 & 2 & 1 \\ 0 & 0 & 0 \\ -1 & -2 & -1 \end{bmatrix}. \tag{5}$$

The effect of this is to take what is highly similar as taking a gradient operator in either direction, with smoothing. There is a correlation coefficient of 0.98 between $G_y$ and the $y$ gradient, and there is a correlation coefficient of 0.999 between $G_x$ and the $x$ gradient. The motivation for using the Sobel operator rather than the gradient operator is principally that it can reduce

the noise in data, as shown by application to photographs (Vincent et al., 2009). Hjelmervik and Hjelmervik (2019) used the magnitude of the $x$ and $y$ Sobel operators, which approximates the magnitude of the gradient, to examine fronts in the Arctic and North Atlantic. They show that the magnitude of the Sobel gradient can be thresholded to highlight features such as the Gulf stream.

Rather than working with the gradient magnitude, Fig. 6 shows $G_y$*PC1, $G_y$*PC2, $G_y$*PC3 alone. This is more inter-

pretable, as the $G_y$*PC1 component is strongly correlated to the zonal velocity $U$ ($r = 0.92$). Hjelmervik and Hjelmervik (2019) use a threshold value to define fronts, but instead we plot the gradient directly as a colormap for each PC, which is useful as it does not obscure any information about the fronts themselves. Appendix A shows that the correlation between the





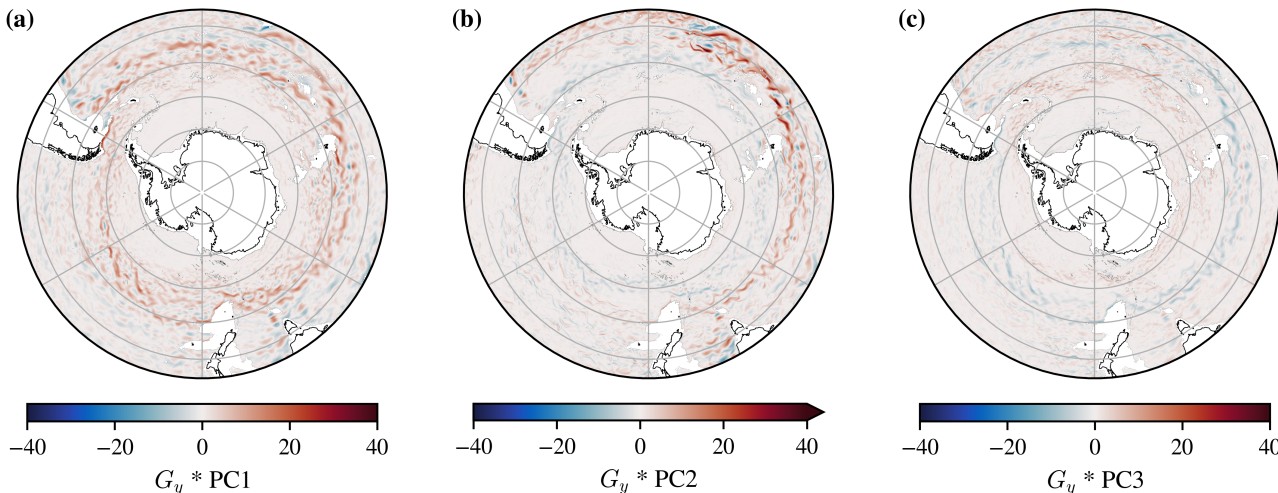

**Figure 6.** $G_y$ Sobel edge operator on PCA for the month of June 2011. The correlation coefficient between panel (a) for PC1, and the zonal velocity $U$ for the same month in B-SOSE-i106 is 0.915, showing that the structure it highlights is substantially similar to the ACC. Panels (b) and (c) for PC2 and PC3 also are also related to the ACC, (correlation coefficients of 0.25, and -0.19 respectively). Grey line is 2000 m isobath.

Sobel $G_y$ gradient of PC1 with the meridional velocity, $V$, and the correlation between $G_x$*PC1 and zonal velocity $U$ increase for roughly the first 2 years of BSOSE-i106, suggesting that the model is still spinning up to geostrophic balance.

The GMM and Sobel methods are complimentary. GMM reveals the large-scale temperature and salinity structure associated with changes in stratification, which has traditionally been used to define the fronts, whereas edge detection methods like the Sobel method used here reveals the smaller-scale structure of multiple jets, which can merge and separate. As such, both approaches may be useful ways of characterising ocean structure without making ad-hoc assumptions related to particular property values or strict requirements that the structures be circumpolar and continuous. The present proliferation of front
definition and analysis methods is driven by the need to expand how the oceanographic community deals with ocean structure across a wide variety of spatial and temporal scales (Chapman et al., 2020).

## 4   Discussion

In this section, we discuss the sensitivity of our results to our choice of dataset (Sect. 4.1), touch on the temporal variability in our results (Sect. 4.2), discuss a possible connecton with the Antarctic Slope Current (Sect. 4.3), examine the sensitivity of
the results to the choice of the maximum number of classes (Sect. 4.4), and discuss the interpretation of posterior probabilities (Sect. 4.5).





### 4.1 Sensitivity to choice of dataset

We chose to use B-SOSE data for this study in order to (1) work with a dataset that features relatively uniform coverage in latitude-longitude and (2) to allow us to examine temporal variability as well as spatial variability. B-SOSE is an observationally-constrained estimate of the hydrographic structure of the Southern Ocean, so it does accurately capture many features of large-scale and mesoscale structure (Verdy and Mazloff, 2017). However, because B-SOSE is a numerical model run, it will no doubt have some biases with respect to observations, particularly on smaller scales. We expect that our results would not change dramatically on basin-wide scales across different state estimate and reanalysis products.

To examine the differences of this bias on the class structure and the structure of the inter-class comparison metric, this study could be repeated with a purely observational dataset such as Argo. One trade-off for such a study would be the fact that observational datasets are relatively sparse in terms of both spatial and temporal coverage relative to a state estimate or other numerical model run. One could attempt to use the same GMM trained on B-SOSE with Argo data, but possible biases between B-SOSE and the Argo dataset could make this challenging. It might be possible to adjust for those biases in the data cleaning and preparation step of the analysis; the standardisation process, which is already a part of the analysis presented here, is a step towards this bias removal and correction that may facilitate comparisons between models and observations. Alternatively, one could attempt to re-train the GMM using Argo data alone. This has been done in other studies, so it should be possible in principle (e.g. (Maze et al., 2017; Jones et al., 2019; Rosso et al., 2020)).

### 4.2 Temporal variability of the fronts

We found that the class structure and boundary positions did not feature large temporal variability with respect to the mean state, but much more work could be done to examine this variability and its connection to the processes that determine thermohaline structure (e.g. surface forcing, subsurface mixing, advection). This is outside the scope of our present study, which is focused on proposing a new metric for identifying and tracking boundaries in Southern Ocean structure.

### 4.3 The Antarctic Slope Current

The Antarctic Slope Current (ASC) that separates warmer open ocean waters from the colder waters on the Antarctic continental shelf is an important component of heat transport in the Southern Ocean. It acts to control the flow of warm water onto the continental shelf and eventually under the floating ice shelves. In a recent paper, Thompson et al. (2020) suggest that if the source of the Antarctic Slope Current (ASC) intersects with the ACC in the Bellingshausen Sea, then the ASC source would be considered a major component of the overturning circulation. In our study, we found a diffuse boundary between classes in the Bellingshausen Sea region, which may be relevant for the physical context of the ASC, which is still under investigation (Fig. 4(b)).





### 4.4 Sensitivity to the maximum number of classes

In this study, we chose $K = 5$ as the number of classes based on sensitivity tests and also based on a priori knowledge. Specifically, previous studies used a front structure with five broad regions, delineated by four fronts, so we might expect a value around $K = 5$ based on this (e.g. Orsi et al. (1995); Pollard et al. (2002); Kim and Orsi (2014)).

Generally, the choice of the maximum number of classes $K$ can be thought of as a way to select models of varying degrees of complexity. Statistical models with lower $K$ values are potentially easier to interpret, only capturing the most dominant structures in the dataset. For example, the probabilistic boundary between the two classes in a $K = 2$ statistical model roughly separates colder, fresher Antarctic waters from the warmer, saltier subtropical waters (Fig. 7(a)). Notably, in this case, the magnitude of the $I$-metric appears to largely decrease as we follow it from the Atlantic and Indian basins and into the Pacific

basin, indicating that the boundary becomes less sharp with longitude. This possibly reflects the fact that the Pacific basin hosts some of the dominant northward export pathways of Subantarctic Mode Water and Antarctic Intermediate Water, consistent with a less sharp transition between polar and subtropical waters (Iudicone et al., 2007; Herraiz-Borreguero and Rintoul, 2011; Jones et al., 2016). A statistical model with $K = 4$ retains most of the features of our analysis with $K = 5$, but the transition region closest to Antarctica in $K = 5$ is no longer present.

The $K = 5$ statistical model we used in this work captures near-Antarctic and circumpolar structure, as well as some subtropical structure. A more complex statistical model with higher $K$ would capture more of the subtropical structure (not shown). This is consistent with sensitivity studies using temperature-only Argo data, where increasing $K$ added details to the subtropical class structure while leaving the circumpolar class structure largely unchanged (Jones et al., 2019). Statistical models with much higher $K$ values may capture more structure in the data, but increasing $K$ also risks overfitting. That is, if we tune

the GMM statistical model to match an increasing number of structures in PC space, we risk losing generality; the goal is to represent the dominant structures of the dataset without overfitting every small variation, some of which could represent noise in the data. This has a direct analogue with overfitting in terms of simple statistical models; it is unwise to use a 10th-order polynomial when a quadratic captures the dominant features of the dataset, because the higher-order polynomial is less likely to generalise to other similar datasets. In addition, statistical models with very high $K$ values are increasingly difficult to interpret

in terms of our current physical and biogeochemical understanding. Note that regional studies, e.g. those carried out in specific sectors of the SO may find it useful to increase $K$ based on local structure (e.g. Rosso et al. (2020)). This is consistent with the suggestion by Chapman et al. (2020) that front definitions may need to be more flexible and region-specific, as opposed to expecting a particular definition to apply globally (or even across a single ocean basin).

### 4.5 Interpreting posterior probabilities

The posterior probabilities returned by a Gaussian mixture model are affected by our choice of $K$. We should be careful not to over-interpret the posterior probabilities as confidences in the correctness of the assigned labels. Notably, GMM does not indicate the probability that a given profile belongs to *none* of the classes in a given statistical model. With that in mind, we can interpret the posterior probability as a measure of unambiguity *in the context of* a given statistical model. When one

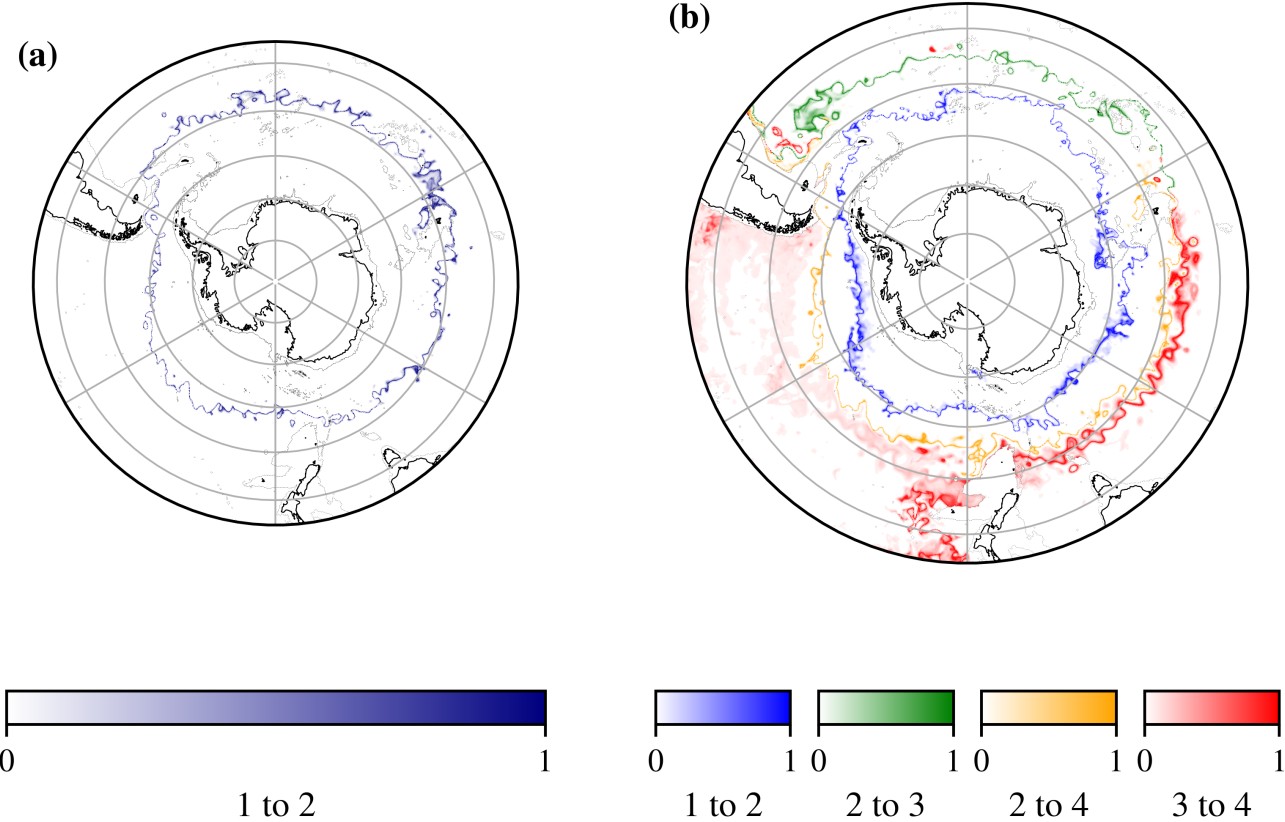

**Figure 7.** Decreasing $K$ removes details from the statistical description of Southern Ocean thermohaline structure. Shown is the GMM-derived $I$-metric, using (a) $K = 2$ and (b) $K = 4$, for a monthly average over June 2011. Grey line is 2000 m isobath.

probability is larger than all others with some margin, the profile is unambiguously classified, while probabilities of similar

magnitude indicate that the profile cannot be unambiguously classified in the current statistical model with the specified number of classes. In this study, we used the posterior probability distribution to identify boundaries between coherent thermohaline regimes, taking advantage of this property of GMM.

## 5   Conclusions

In this study, we proposed a new metric for defining and identifying boundaries between coherent regimes of temperature

and salinity structure. Our method uses Gaussian mixture modelling, a type of unsupervised machine learning, to establish a statistical model of thermohaline structure that is intended to capture the large-scale features of the dataset in both PC space and in geographic space. We developed our method in the Southern Ocean due to the presence of circumpolar structures and





relatively clear fronts, but our approach could be applied to other regions or even to the global ocean as a whole. The *I*-metric
provides a flexible, probabilistic method to define and identify boundaries in an oceanographic dataset without using ad hoc
property contours; the boundaries are derived in a generalised method that reflects the structure of the dataset. The *I*-metric has
potential as a method for comparing different observational and numerical modelling datasets in a robust, algorithmic way that
is not heavily affected by biases in the mean state between datasets. It features a parameter $K$ that allows users to increase and
decrease the level of complexity of the statistical model; the optimal value of $K$ will vary between applications. As discussed
in Chapman et al. (2020), the field of oceanography needs to consider fronts and boundaries in a more general, application-
specific way, due in part to the richness of ocean structure on different spatial scales. The *I*-metric was designed with this
problem in mind; it is intended to be a complementary addition to the oceanographic toolbox as opposed to a replacement for
any particular method.

*Code and data availability.* B-SOSE iteration 106 state estimate data is available from the Scripps Institution of Oceanography (http://
sose.ucsd.edu/BSOSE6_iter106_solution.html). The MITgcm source code that was used to create B-SOSE is available on GitHub (https:
//github.com/MITgcm/MITgcm). Original climatological front positions from Kim and Orsi (2014) are available on Researchgate (https:
//www.researchgate.net/publication/338420242_ACC_fronts). The code used to carry out the analysis and figure creation for this paper
is available via Zenodo (Thomas, 2021) (Up-to-date repository: https://github.com/so-wise/so-fronts). This software uses scikit-learn as a
foundation (Pedregosa et al., 2011). We used Cartopy for mapping (Met Office, 2010 - 2015).





## Appendix A: The relation between edge detection and the velocity field

### A1 Fronts and jets

Here we discuss fronts and jets (Cushman-Roisin and Beckers, 2011, chapters 15 and 18). If we first assume that the flow is steady and hydrostatic, and that pressure gradient force equal the Coriolis force (i.e. geostrophic balance) then:

$$-fv = -\frac{1}{\rho_0}\frac{\partial P}{\partial x} \tag{A1}$$

where $\rho_0$ is the reference density, assuming a Boussinesq approximation, $P$ is the pressure, $x$ is the distance along the x-axis, $v$ is the component of velocity along the y-axis, and $f$ is the Coriolis parameter. This means that

$$\frac{\partial v}{\partial z} = -\frac{g}{\rho_0 f}\frac{\partial \rho}{\partial x}, \tag{A2}$$

and given that we the hydrostatic relation

$$\frac{\partial P}{\partial z} = -\rho g, \tag{A3}$$

we can therefore write

$$-f\frac{\partial v}{\partial z} = -\frac{1}{\rho_0}\frac{\partial}{\partial x}\frac{\partial P}{\partial z} = -\frac{1}{\rho_0}\frac{\partial}{\partial x}\left(-\rho g\right), \tag{A4}$$

which means that in the final form,

$$\frac{\partial v}{\partial z} = -\frac{g}{\rho_0 f}\frac{\partial \rho}{\partial x}. \tag{A5}$$

Equivalently in the perpendicular direction

$$\frac{\partial u}{\partial z} = \frac{g}{\rho_0 f}\frac{\partial \rho}{\partial y}. \tag{A6}$$

(Note the change in sign between the two directions between Eq.s A5 and A6.) The vertical shear of zonal velocity is related to the meridional gradient of density; sharp density changes can correspond to strong geostrophic jets by this balance.

### A2 Edge detection and velocity field

Fig. A1 and Fig. A3 illustrate the spatial resemblance between $G_y$*PC1 and $U$, $G_x$*PC1 and $V$, respectively, compared over June 2011 or as an average over the full B-SOSE period. The domain-averaged correlation is shown quantitatively in Fig. A2 and Fig. A4, where in the last couple of years of the reanalysis product there is an especially high correlation between the two. That Fig. A2 and Fig. A4 show opposite signs in the correlation is equivalent to the reversal in sign between Eq.s A6 and A5. Those figures also show that the magnitude of the correlation between the respective variables increases during the first two years of the dataset before flattening off. This is suggestive of the model spinning up towards geostropic balance. As the first principal component statistically explains the first order structure in the ocean, it primarily represents the density contrast produced by the thermohaline structure from the tropics to the poles.





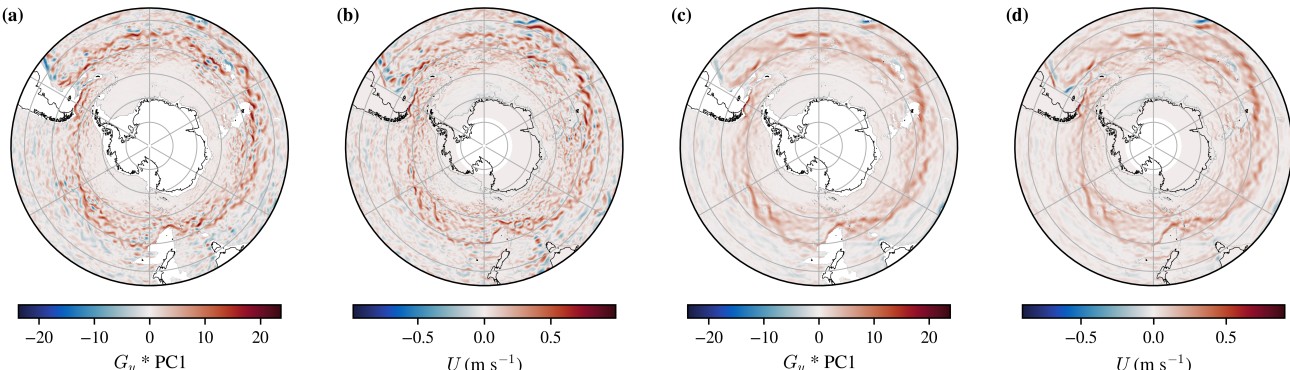

**Figure A1.** A comparison between the $G_x$*PC1 Sobel edge detection field and the zonal velocity, $U$, at 135m. Panels (a) and (b) are from monthly mean over June 2011, whereas (c) and (d) are the mean over all of the monthly means in the dataset.

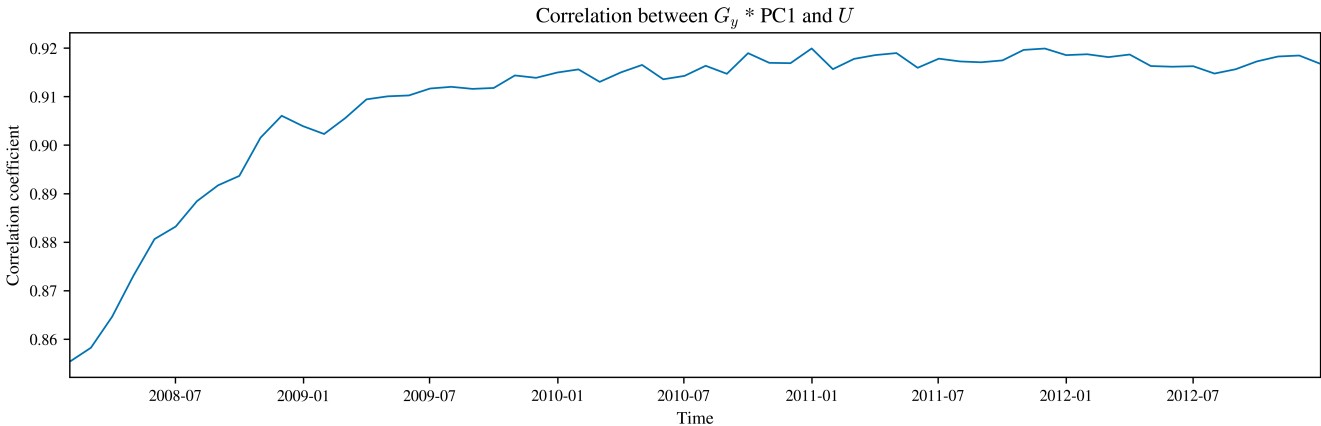

**Figure A2.** The correlation between $G_x$*PC1 and the zonal velocity, $U$, at 135m, for each monthly mean in the BSOSE-i106 dataset. The increase in the correlation over the first two years could be interpreted as the spin up.


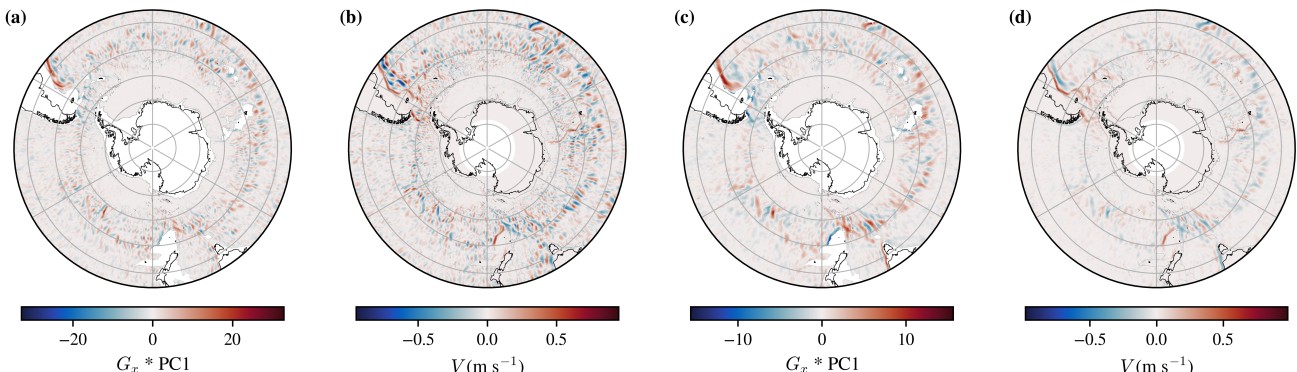

**Figure A3.** A comparison between the $G_y$*PC1 Sobel edge detection field and the meridional velocity, $V$, at 135m. Panels (a) and (b) are from monthly mean over June 2011, whereas (c) and (d) are the mean over all of the monthly means in the dataset.

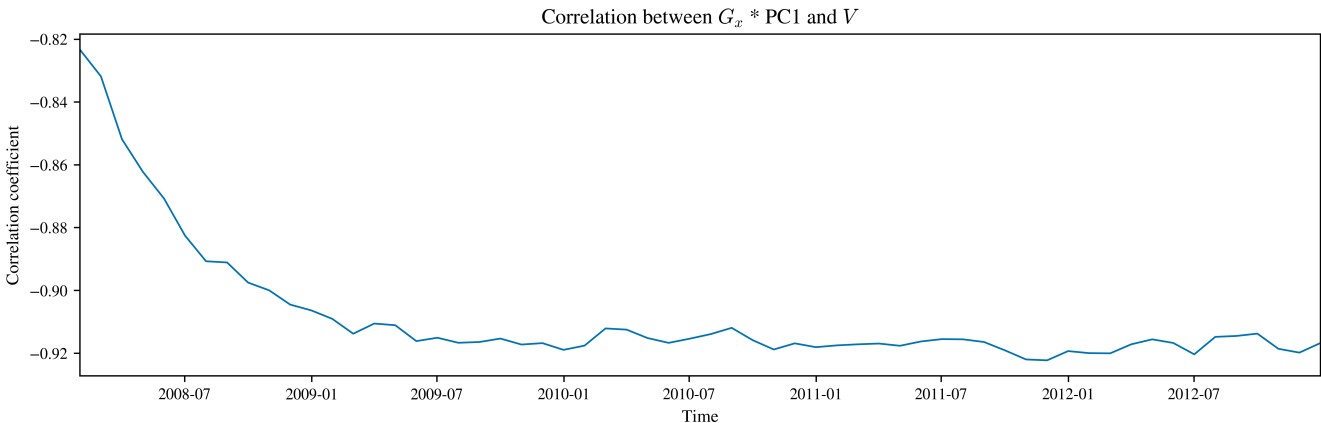

**Figure A4.** The correlation between the $G_y$*PC1 and the meridional velocity, $V$, at 135m for each monthly mean in the BSOSE-i106 dataset. The increase in the correlation coefficient over the first two years could be interpreted as the spin up as in Fig. A2





## Appendix B: Gaussian mixture modelling

A Gaussian mixture model (GMM) attempts to represent a dataset using a linear combination of multi-dimensional Gaussian distributions. A multi-dimensional Gaussian (Eq. B1), is a simple generalisation of a Gaussian to $D$ dimensions. In this case, the dimensions correspond to the three principal components of the data.

$$\mathcal{N}\left(\boldsymbol{x}_n;\boldsymbol{\mu}_k,\Sigma_k\right)=\frac{\exp\left[-\frac{1}{2}\left(\boldsymbol{x}_n-\boldsymbol{\mu}_k\right)^T\left(\Sigma_k^{-1}\right)\left(\boldsymbol{x}_n-\boldsymbol{\mu}_k\right)\right]}{\sqrt{(2\pi)^D\,\|\Sigma_k\|}}. \tag{B1}$$

We make the assumption that the probability distribution that generated the dataset can be approximated by a set of multivariate Gaussians (Eq. B2):

$$\widetilde{\mathbb{P}}(\boldsymbol{x}_n)\simeq\mathbb{P}(\boldsymbol{x}_n)=\sum_{k=1}^{K}\lambda_k\,\mathcal{N}\left(\boldsymbol{x}_n\,;\,\boldsymbol{\mu}_k\,,\,\Sigma_k\right). \tag{B2}$$

Any probability distribution function (PDF) could be described by an arbitrarily large number of Gaussians (Eq. B3), but to be a good method of describing the data this should be a manageable number.

$$\widetilde{\mathbb{P}}(\boldsymbol{x}_n)=\lim_{K\to\infty}\sum_{k=1}^{K}\lambda_k\,\mathcal{N}\left(\boldsymbol{x}_n\,;\,\boldsymbol{\mu}_k\,,\,\Sigma_k\right) \tag{B3}$$

In this paper, we showed that our Southern Ocean thermohaline dataset can be fairly represented as a series of plateau-like regions in PC variable space, so it can be approximated by a PDF made from a set of multivariate Gaussians, where the boundaries between these Gaussians correspond to the fronts (Fig. 2).

## B1 Expectation Maximisation

To initialize the method, first $K$ clusters are created randomly, or perhaps from a $k$-means clustering application. Next, the set of Gaussians is iteratively adjusted (Eq.s B4, B5 and B6) until it reaches a local minimum in the cost function Maze et al. (2017). It is generally expected that reducing the number of dimensions in the pre-processing step helps improve the convergence. The following section draws heavily from Maze et al. (2017).

$$\lambda_k{}^{(t+1)}=\frac{1}{N}\sum_{n=0}^{N-1}\mathbb{P}\left(c_n=k\mid\boldsymbol{x}_n\,;\,\{\lambda_k\,,\,\boldsymbol{\mu}_k\,,\,\Sigma_k\}^{(t)}\right) \tag{B4}$$

$$\boldsymbol{\mu}_k{}^{(t+1)}=\frac{\sum_{n=0}^{N-1}\mathbb{P}\left(c_n=k\mid\boldsymbol{x}_n\,;\,\{\lambda_k\,,\,\boldsymbol{\mu}_k\,,\,\Sigma_k\}^{(t)}\right)\boldsymbol{x}_n}{\sum_{n=0}^{N-1}\mathbb{P}\left(c_n=k\mid\boldsymbol{x}_n\,;\,\{\lambda_k\,,\,\boldsymbol{\mu}_k\,,\,\Sigma_k\}^{(t)}\right)} \tag{B5}$$

$$\begin{aligned}\Sigma^{k(t+1)}=\quad&\sum_{n=1}^{N}\mathbb{P}\left(c^n=k\mid\boldsymbol{x}^n;\,\{\lambda^k,\,\underline{\mu}^k,\,\Sigma^k\}^{(t)}\right)\\&\cdot\left(\boldsymbol{x}^n-\boldsymbol{\mu}^{k+1}\right)\left(\boldsymbol{x}^n-\boldsymbol{\mu}^{k+1}\right)^T\\&\div\sum_{n=0}^{N-1}\mathbb{P}\left(c^n=k\mid\boldsymbol{x}^n;\,\{\lambda^k,\,\boldsymbol{\mu}^k,\,\Sigma^k\}^{(t)}\right)\end{aligned} \tag{B6}$$





## B2   Information Criterion

GMM needs an input hyperparameter $K$ that sets the number of clusters that will be fitted to the data. GMM is relatively cheap
to run, and so it is reasonable to run it with a large range of $K$ and choose the $K$ which best describes them. The often used
criterions are the Bayesian Information Criterion (BIC) (Eq. B7) and Akaike Information Criterion (AIC) (Eq. B8). They both
essentially contain a term which measures the agreement of the model to the data, and have a penalty term for the number
of parameters that have been used to achieve this (related to $K$). So we are looking for a minima in AIC / BIC to guide our
choice of $K$. There is no clear minimum for this dataset in $K$ for $2 \leq K \leq 100$, which is typical of oceanographic applications
due in part to the highly correlated nature of the data (e.g. (Sonnewald et al., 2019; Jones et al., 2019)). Because $K$ is weakly
constrained, we are able to select a lower value of $K$ for ease of interpretation, having verified that it captures the large-scale
structure of the data in PC space, which is suitable for our purposes. BIC and AIC take the forms:

$$\text{BIC} = -2\mathcal{L}(K) + \eta_f(k)\log(N) \tag{B7}$$
$$w. \quad \eta_f = K - 1 + KD + \frac{KD(D-1)}{2}$$
$$\text{AIC} = 2K - 2\mathcal{L}, \tag{B8}$$

where the log-likelihood is expressed as:

$$\mathcal{L} = \log[\mathbb{P}(X)] = \sum_{n=0}^{N-1} \log\left(\sum_{k=1}^{K} \lambda_k \, \mathcal{N}\left(\boldsymbol{x}_n \, ; \, \lambda_k \, , \, \boldsymbol{\mu}_k \, , \, \Sigma_k\right)\right) \tag{B9}$$

## B3   Labelling the Data Set

Each data point is assigned a posterior probability distribution across the $K$ clusters (Eq. B10). This uncertainty information is
one of the useful features of GMM. The probability takes the form:

$$\mathbb{P}\left(c_n = k \mid \boldsymbol{x}_n \, ; \, \lambda_k \, , \, \boldsymbol{\mu}_k \, , \, \Sigma_k\right) = \frac{\lambda_k \, \mathcal{N}\left(\boldsymbol{x}_n \, ; \, \boldsymbol{\mu}_k \, , \, \sum_k\right)}{\sum_{k=0}^{K-1} \lambda_k \, \mathcal{N}\left(\boldsymbol{x}_n \, ; \, \boldsymbol{\mu}_k \, , \, \sum_k\right)}. \tag{B10}$$

To label a dataset, each data point is assigned a label from the cluster that it would be the most likely to be generated by, in a
statistical sense (Eq. B11).

$$\mathcal{C} = \arg\max_k \left(\mathbb{P}\left(c_n = k \mid \boldsymbol{x}_n \, ; \, \lambda_k \, , \, \boldsymbol{\mu}_k \, , \, \Sigma_k\right), \, 1:k\right) \tag{B11}$$



*Author contributions.* DJ designed the initial project, and both ST and DJ developed it further. ST wrote the software (Thomas, 2021), performed the analysis, and created the figures. AF proposed a significant improvement to the inter-class metric. EM and EP provided expert guidance on fronts, Southern Ocean structure, and dynamics. ST and DJ wrote the initial manuscript, and all authors assisted with edits.

*Competing interests.* No competing interests

*Acknowledgements.* This work originated as a Natural Environment Research Council (NERC) Research Experience Placement (REP) project funded by the SPITFIRE Doctoral Training Partnership (reference NE/S007210/1). DJ is supported by a UKRI Future Leaders Fellowship (reference MR/T020822/1). DJ and ST also received funding from the NERC ACSIS project (reference NE/N018028/1). EP received funding from the European Research Council (ERC) under the European Union's Horizon 2020 research and innovation program
(Grant Agreement 637770). The authors would like to thank Emma Boland, Peter Haynes, Guillaume Maze, and John Taylor for comments that improved the quality of this work.



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
