# Peer review of "Defining Southern Ocean fronts using unsupervised classification"

_Ocean Science, 2021_

## Author Comment (AC1)

**Response to Reviewer 1**

Thank you for your comments on our paper. Below, please see our responses (in black) to the comments (in blue). We hope that you find our manuscript to be improved and suitable for publication.

The authors developed a new method for detecting oceanographic fronts by employing PCA, GMM, and I-metric. By utilizing B-SOSE, they have demonstrated that their new methods successfully detect fronts, which vary in time and space. In general, I think their scientific discussions are sound. I think this manuscript is well-written. However, for readers who do not know much about these analyses methods, some parts of the text are difficult to follow. I suggest minor revision before publication.

Thank you for these comments, we have endeavoured to improve the manuscript in response to these recommendations.

Minor comments

Section 2.1 For those people who do not know much about ECCO, it would be great if authors can elaborate a bit about what other ECCO products are available and why authors choose B-SOSE.

Good suggestion, thank you. That has been added.

Lines 117-126: This part of the text is difficult to follow without reading Pauthenet et al., 2017 carefully. A bit more explanation (e.g., equations (1) in Pauthenet et al., 2017) and something similar to Fig. 4 in Pauthenet et al., 2017 would be helpful for better understanding.

Thank you for pointing this out. We have added additional detail to section 2.2, and we have added an appendix (Appendix B) with our version of Fig. 4 from Pauthenet et al. (2017) (Figure B2). We hope that this helps clarify our procedure.

Line 147-170: If I am not wrong, I think PCA is used just to compress the data (reduce its number of dimensions), rather than to investigate the structure of its covariance matrix. This point is not stressed in this paper. The authors should clarify this point if this is correct.

The PCA is not just used to compress the data, we also look at the PC spatial distribution in figure 1, which helps us understand the main modes of the thermohaline structure.

In addition, it becomes easier to understand if the authors can elaborate on (1) why GMM is used rather than PCA (advantages and disadvantages) and (2) the difference (or relations)

between vertical modes of temperature and salinity obtained from PCA analysis and figure 5. For me, this manuscript was initially very confusing because it took me a while to understand that PCA is only used to compress the data but main analysis was conducted using GMM.

Thank you for these comments, that is helpful. I have tried to add an additional sentences to prime the reader with this purpose.

Abstract and Summary :

Authors conduct analysis using Sobel edge operator but they do not argue how it can be useful in the abstract and conclusion. A sentence describing how it can be useful and comparisons with GMM is going to be useful.

Thank you for this comment, we have added more comments to the abstract (lines 15-19), summary, and conclusions (lines 396-397). Specifically, we mention that Sobel edge detection may be useful for defining and tracking smaller-scale fronts in model and reanalysis data.

---

## Author Comment (AC2)

**Response to Reviewer 2**

Thank you for your positive and constructive comments on our paper. Below, please see our responses (in black) to the comments (in blue). We hope that you find our manuscript to be improved and suitable for publication.

The authors propose a new method to quantitatively define the Southern Ocean fronts. The new metric, I-metric, searches boundaries of water masses in the space spanned by the principal components of the Southern Ocean hydrography. The water masses are classified by the Gaussian mixture modelling. When applied to the output of the Southern Ocean State Estimate (B-SOSE), the newly defined fronts correspond well to the traditional definitions of fronts both in monthly snapshots and in 5-year average.

I find the definition of the I-metric (equation (3)) is intuitive and natural, and am convinced that the new method has an advantage over the traditional front definitions in its ability to allow "for a wider variety of transition types between regimes" (ll.198-199). The paper is well written except for a few places where exact meanings of technical terms are not obvious (see below). I think only a minor revision is needed before acceptance.

Thank you for these comments, they were very helpful.

I.95: Doesn't B-SOSE also solve the salt conservation equation?

Yes, thank you for catching this omission. Added (line 106).

Table 1. It is stated that "a state estimate ... the changes in ... mixing parameter" (l.90). Are the mixing parameters listed in Table 1 the initial values? Or are they fixed? Or are they after adjustment?

It is true that mixing parameters are varied in some ECCO products, but in B-SOSE the parameters in Table 1 are manually adjusted and held fixed. We have changed the text to better reflect this procedure.

I.120: Explain the use of the word "normalise" in more detail. Did you calculate the spatial standard deviation at each depth at each time step in the entire region south of 30S and divide temperature with the standard deviation? Or the standard deviation includes the time dimension?

The standard deviation and the mean are calculated at each point for each variable at each depth level, over the full sample year. I have added additional sentences at this point to clarify this point. We have also added a couple of plots in a new appendix (now called appendix B) to illustrate the preprocessing steps. (line 418-420)

l.180: I believe this term "posterior probability" is used in the Bayesian probability. If this is the case, it would be helpful to explain what is the prior probability here.

We have added one line to the manuscript (at line 187) to make this point. We use the term posterior probability because it is used in other papers that use GMM, and so we use this term for consistency.

ll.183, 190, etc. also equation (3):  Is "maximum" right word here? The word "highest" (l.185) sounds more appropriate since all Gaussian are "maximum" (l.165) in the sense of minimum mode-data misfit.

I agree that this seems to be a more clear wording. Thank you for this suggestion. All instances have been changed to "highest" from "maximum".

l.246:  By "central" and "eastern" export pathways, I interpret the former corresponds to the equatorward path around the dateline and the latter along the 120W meridian. If the "eastern" means the I-metric blob off the South America coast, more detailed explanation is needed to clarify the ambiguity.

Thank you for pointing out this ambiguity. The export pathways spatially correspond to the *low* values of the *I*-metric; the higher values delimit the boundary of the pathways. We have changed the text to hopefully remove this ambiguity.

ll.277-278: I do not understand what is meant by "correlation coefficient between Gx and x gradient." Is this the correlation between the result of Gx operator and the result of d/dx applied to the gridded PC data?

Yes, this was a comparison between applying the Gx operator and the d/dx operator, to the same PC1 grid. We have attempted to clarify this (line 296-298)

l.283: What is "*" operator? How is it defined? Convolution in space?

Yes, this is defined as the 2d convolution in space (line 272). I have added this to the figure caption as well to make it more obvious for the reader.

Equations (B4)(B5)(B6): $c_n$ seems undefined. mu in the 1st line of (B6) should be in bold face.

We have added that definition to the start of the definition, and made sure that mu is always bold (line 432, 437, 440).

Equation (B7): Is k defined?

We have added the definition to the beginning of appendix A (line 433), thank you for pointing this out.

L.450: What is w here?

"w." was an abbreviation for "with". Thank you for catching this. I have replaced this with "with" (line 467).

Appendix A1.

(A2) and (A5) are identical. The content is very elementary and can be found in any textbook. I do not think this appendix is necessary. I found Appendix A2 is very informative.

Thank you, it is very helpful to hear what you found to be useful background, and what is unnecessary. We deleted appendix A1, but kept a reference to the appropriate textbook, in case students need to be guided to the right resource. We have left the second subsection A2 much as it was, and are glad that you found it helpful.

---

## Author Comment (AC3)

**Response to Reviewer 3**

This paper proposes a concise and effective metric for identifying ocean fronts. The authors provide thorough background and reasonable discussion of certain considerations when using the metric.

Thank you for your comments and consideration. Below, please see our responses (in black) to the comments (in blue). We hope that you find our manuscript to be improved and suitable for publication.

Comments:

While the metric does appear to effectively highlight when water masses are not clearly assigned to a single cluster – the authors remain ambiguous on what is a 'low' metric and what is a 'high' metric. Toward the goal of removing ad hoc decision making, a more quantitative description of how the I-metric is classified would be helpful.

The difference between "low" and "high" is indeed somewhat arbitrary. In any particular application, one could use a histogram approach to find potential cut-off values. Here there is a sharp contrast between open ocean values and along-front values, which makes the separation a bit clearer.

Similarly, in figure 2b, low I-metric points are not shown – what was the threshold used and how was it chosen?

The threshold value was chosen at 0.05. This threshold is quite subjective but was chosen to improve the visibility of the plot. It also corresponds to standard values chosen for statistical significance.

Figure 3 caption should be rephrased to read as complete sentences.

This has been changed, and we hope that you now find it more readable.

Figure 5 legend should be fixed to be easily readable.

We changed the figure dimensions and the legend to be more readable. The lines in the figure legend from Kim and Orsi (2014) altimetric fronts are now much thicker, which should help a reader differentiate between them.

Figure 5 caption should be rephrased somehow, e.g. "Envelope indicates one standard deviation to either side."

Thank you for that, we have changed that (last sentence in caption of Figure 5).

First sentence after equation 5 (line 277) should be rephrased more clearly.

Modified for clarity (line 296).